# Burden of COVID-19 on Italian Internal Medicine Wards: Delphi, SWOT, and Performance Analysis after Two Pandemic Waves in the Local Health Authority “Roma 6” Hospital Structures

**DOI:** 10.3390/ijerph18115999

**Published:** 2021-06-03

**Authors:** Filomena Pietrantonio, Francesco Rosiello, Elena Alessi, Matteo Pascucci, Marianna Rainone, Enrica Cipriano, Alessandra Di Berardino, Antonio Vinci, Matteo Ruggeri, Serafino Ricci

**Affiliations:** 1Internal Medicine Unit, Castelli Hospital, Azienda Sanitaria Locale Roma 6, 00072 Ariccia, Italy; filomena.pietrantonio@gmail.com (F.P.); elena.alessi@aslroma6.it (E.A.); matteo.pascucci@aslroma6.it (M.P.); marianna.rainone@aslroma6.it (M.R.); enrica.cipriano@aslroma6.it (E.C.); alessandra.diberardino@aslroma6.it (A.D.B.); 2Department of Health Economics, St. Camillus University Health and Medical Sciences, 00131 Rome, Italy; mruggeri76@gmail.com; 3Department of Public Health and Infectious Disease, Sapienza University of Rome, 00185 Rome, Italy; 4School of Specialization in Hygiene and Preventive Medicine, University of Rome “Tor Vergata”, 00133 Rome, Italy; antonio.vinci.at@hotmail.it; 5National Centre for HTA, National Institute for Health, 00161 Rome, Italy; 6Department of Anatomical, Hystological Sciences and Legal Medicine, Sapienza University of Rome, 00185 Rome, Italy; serafino.ricci@uniroma1.it

**Keywords:** COVID-19, Internal Medicine ward, territorial medicine

## Abstract

*Background*: COVID-19 causes major changes in day-to-day hospital activity due to its epidemiological characteristics and the clinical challenges it poses, especially in internal medicine wards. Therefore, it is necessary to understand and manage all of the implicated factors in order to maintain a high standard of care, even in sub-par circumstances. *Methods*: This was a three-phase, mixed-design study. Initially, the Delphi method allowed us to analyze the causes of poor outcomes in a cohort of an aggregate of Italian COVID-19 wards via an Ishikawa diagram. Then, for each retrieved item, a score was assigned according to a pros/cons, opportunities/threats system. Scores were also assigned according to potential value/perceived risk. Finally, the performances of MCs (Medicine-COVID-19 wards) and MCFs (Medicine-COVID-19-free: Internal Medicine wards) units were represented via a Barber’s nomogram. *Results*: MCFs hospitalized 790 patients (−23.90% compared to 2019 Internal Medicine admissions). The main risk factors for mortality were patients admitted from local facilities (+7%) and the presence of comorbidities (>3: 100%, ≥5: 24.7%). A total of 197 (25%) patients were treated with non-invasive ventilation (NIV). The most deaths (57.14%) occurred in patients admitted from local facilities. *Conclusions*: Medicine-COVID-19 wards show higher complexity and demand compared to non-COVID-19 ones and they are comparable to sub-intensive therapy wards. It is necessary to promote the use of NIV in such settings.

## 1. Introduction

Shortly after the beginning of the COVID-19 pandemic (03/11/2020) [1,2,3], and for the entire duration of the outbreak, Italy has been among the world’s worst-hit countries, to the point of being the first Western country to ever undergo a national lockdown [4].

At the time of writing this manuscript, the number of reported cases is still among the highest in Europe [5]. Since the beginning of the pandemic, the number of cases in the country has been over 4,130,000, with, unfortunately, more than 122,000 deaths [6,7,8,9,10]. The reason for this public health failure was mainly related to a fragmentation of the decision-making process, with a lack of a strong, nationally centralized response to the emergency [11]. This is particularly evident when observing the different regional policies implemented, i.e., in terms of testing strategies, exacerbated by years of public sector neglect, with an increase in out-of-pocket expenditures and public healthcare budget cuts, and perpetrated by governments of all political affiliations [12].

Currently, there are two main alternatives for the hospitalization of symptomatic SARS-CoV-2 patients: Internal Medicine wards and Intensive Care (IT) wards. The choice of destination is made by considering the seriousness of the clinical presentation and the patient’s comorbidities [13].

Internal Medicine wards are regarded as “high complexity” wards, while intensive care wards are classified as “high intensity” wards and are characterized by a higher death rate. For the entire system [14,15], costs for a typical COVID-19 patient are much higher in terms of both direct costs (included diagnostics, treatment and consultancy) and indirect costs related to the sheer length of recovery.

Since the end of 2020, at least one vaccine has been made available throughout Europe [16], and the vaccination campaign in Italy has begun accordingly [17].

### 1.1. The Local Health Authority “Azienda ASL Roma 6”

Azienda Sanitaria Locale (ASL) Roma 6 is a Public Health Authority in the Lazio Region in Central Italy. It is staffed with 3312 employees (including administration and healthcare providers) [18]. It includes in its territory 21 municipalities (658 km^2^) (Figure 1); it is functionally distributed into six districts (numbered H1–H6) according to the demographic and orographic characteristics of the territory; and it serves a total population of 574,976, whose demographic distribution by age and gender is similar to Italy’s overall population [19].

It also includes 4 hospital structures (PO–*Presidio Ospedaliero*), 8 medium-intensity private structures (working under an accreditation regime, with partial public commissioning, funding and supervision), and at least 36 low-intensity territorial facilities (RSA—*Residenza Sanitaria Assistita*) for vulnerable patients (elderly, psychomotor-impared people, etc.). After the COVID-19 syndemia (20), ASL Roma 6 implemented a COVID-19 hub hospital (*Nuovo Ospedale dei Castelli Romani*, NOC) and two COVID-19 RSA hospitals (the former PO of *Genzano di Roma*, previously closed in 2016, and the PO of *Albano Laziale*, previously closed in 2018). As of 12/31/2020, according to phase nine of ASL’s facilities rearrangement, and based on the COVID-19 Emergency Plan drafted by the administration of the Lazio Regional Government, the number of available beds at NOC was, respectively, 91 in the COVID-19-free Internal Medicine (MCF) Department (including wards of Cardiology, Psychiatry, Pediatric and Gynecology, and the Internal Medicine ward, which was not always active), and 205 in the COVID-Internal Medicine (MC) Department (including the wards of Multidisciplinary Surgery, Intensive Care, Emergency Medicine, and COVID-19 Internal Medicine). The 24 beds already available in the Internal Medicine ward, as of the year 2019, were converted to MC in April 2020, increased to 37 in May, to 38 in October, to 129 in November, and lastly reduced to 91 in December.

Meanwhile, in the PO of the *Albano Laziale* facility, 29 beds were transferred, which were already included in the NOC hospital’s MC count.

According to the syndemia [20] trend in Italy, the Roma 6 territory experienced two COVID-19 waves in 2020: the first one during spring 2020 (February–June) and the second one between October and December 2020, with a total of 354 COVID-19 patients. ASL Roma 6 represented, therefore, a typical Italian Health Authority, whose structuration, facilities, and territories shared the same basic structure across the country [18], in terms of both population and organization. This, combined with data availability, represented the reason for it being chosen by the authors as a case scenario for studying the burden of hospital wards during the COVID-19 pandemic in Italy.

## 2. Materials and Methods

### 2.1. Study Design

Three-phase mixed design.

*Phase 1*: Delphi Method application [21].

The Delphi method is a forecasting process framework based on the results of multiple rounds of questionnaires sent to a panel of experts. Its application is quick and easy and can lead to a consensus among a group of experts. In this case, the personnel involved in the MC and MCF wards took turns to answer the question: “*What is the reason behind bad clinical outcome in some COVID patients?*”

*Phase 2:* Strengths, Weaknesses, Opportunities, and Threats (SWOT) analysis [22,23].

For any topic retrieved during the previous phase, a panel of experts (10 people, with clinical, public health, health economics, and statistics expertise) assigned a score relating to:
−perceived value resulting from the balance between strengths and weaknesses;−potential risk resulting from the balance between threats and opportunities.
These scores ranged on a Likert scale from -10 (minimum added value or minimum risk) to +10 (which corresponded to the maximum added value or maximum risk) [9]. Then, a scatterplot graph was drawn. Overall weights relating to the perceived value were plotted on the *x*-axis and the associated potential risk was plotted on the *y*-axis. The risk/value ratio could fall into any of the following 4 areas, as identified by the graph: a low risk and a high value area (Comfort Zone), a high risk and low value area (Danger Zone), a high risk and high value area (Challenge Area), and a low risk and low value area (Improvement Area).

*Phase 3*: Retrospective Ecological Performance Analysis (with data retrieved from existing sources).

During this phase, data from the hospital discharge forms (*SDO—Scheda Dimissione Ospedaliera*) of the patients were retrieved and analyzed.Descriptive population statistics were calculated (age, death rate, number of comorbidities, and provenience). Likewise, for every ward unit involved in the study, all of the elements necessary for drawing a Barber’s nomogram were either retrieved (total patient numbers, total admission days, total admission numbers, total discharges/deaths, start/end of activity dates, and available bed numbers) or calculated (average bed availability per month (B/M), bed occupancy rate (BOR), length of stay (LOS), turnover index (TI), bed rotation index (BRI), and admitted patients’ average age (AV.AG)).Barber’s nomogram is an index of bed utilization and is defined on the basis of the following: LOS [24], average LOS [25], BOR [26], TI [27], and BRI. A nomogram that takes into account the aforementioned variables, and allows the setup of a range of values in order to assess the performance in the delivery of care, is then derived. In our study, the following was considered acceptable: a LOS within 12 days, a BOR of 75%, and a maximum TI between 1 and 3 days [28].

### 2.2. Population, Variables, and Data Sources

Phase 3 included a population that consisted of all patients admitted to any MCF ward from 1/1/2019 to 31/12/2020, and to any MC from 4/1/2020 to 6/3/2020, and from 10/18/2020 to 31/12/2020. MC dates coincided with the COVID-19 wards’ activation periods in the local Lazio region during first and second waves of the pandemic, respectively; COVID-19 wards were not active during other periods.

MCF and MC wards were active in 2 different hospital structures: PO Albano Laziale and NOC Hospital. As described in Section 1.1, they are both public structures managed directly by the ASL Roma 6 Authority.

The included variables were gender, age, presence of comorbidities, admission date, discharge date, provenience, and clinical outcome.

The data were retrieved from the SDO databank that originated in any of the Roma 6 hospitals.

## 3. Results

### 3.1. Phase 1

In the first stage, the Delphi method [13] was applied to realize an Ishikawa model [29] focused on the reasons behind the deaths of the patients in the COVID-19 wards. Such reasons could be described as belonging to four major groups as follows: (Figure 2).

Human activities (i.e., mostly reasons associated with staff empowerment, motivation, and well-being)Methods (i.e., mostly related to workflow, work mentality, resistance to change)Materials (i.e., situational causes due to material shortages or procedural delays)Environment (i.e., situational causes due to either the emergent pandemic itself, or the structural situation of healthcare in the region, whose manpower and resources have been depleted after years of public sector neglect, healthcare budget cuts, and turnover lock).

### 3.2. Phase 2

For each depicted area, a panel of 10 experts from different fields assigned a different score for risk (defined as the ratio between risk and opportunities) and value (defined as the ratio between weaknesses and strengths) in order to identify what was most in need of intervention. Mean values were then calculated and plotted on a Cartesian graph, in order to depict the SWOT chart. No elements lied in the Danger Zone; problems regarding human activities were, however, perceived as a weakness in the organization and therefore in need of immediate intervention. Lastly, environmental changes were considered as an opportunity for bettering the service (Figure 3).

### 3.3. Phase 3

On the basis of the average availability of beds (per month), the two waves of the pandemic have been considered both separately and jointly. In order to compare the data collected in the MCF ward, years 2019 and 2020 have been considered separately.

As a result of the SDO analysis, the following data were obtained:

MCF 2019: average number of beds per month: 29.25; BOR 86.11%, average hospitalization length 8.8 days, TI 1.42 days, and BRI 35.59%. In 365 days, 1041 people were hospitalized (496 M, 542 F) with a total number of 9168 days of hospitalization and an average patient age of 79.03 years (78.19 M, 79.81 F). Overall, 100% of patients had ≥3 comorbidities. The death rate was 17.38% (181 patients; 89 M, 93 F), the patients’ average age was 86.23 years (85.01 M and 87.43 F), and the average hospitalization length was 6.94 days (min 0, max 30). No data about provenience (home, low intensity facilities, or other) could be retrieved from Emergency Department (ED) discharge reports.

MCF 2020: average number of beds per month: 26.50; BOR 98.98%, average hospitalization length 8.26 days, TI 0.09 days, and BRI 29.81%. In 250 days, 791 people were hospitalized (394 M, 397 F) with a total number of 6531 days of hospitalization and an average patient age of 77.73 years (75.45 M, 80.02 F). Overall, 100% of patients had ≥3 comorbidities. The death rate was 17.38% (117 patients; 58 M, 59 F), the patients’ average age was 82.4 years (82.94 M and 81.86 F), and the average hospitalization length was 6.68 days (min 0, max 41). A total of 46 patients that died in the MCF came from RSA (39.31%), 23 (19.65%) from home, and 45 (38.46%) were unknown.

MC I Wave: average number of beds per month: 30.50; BOR 58.34%, average hospitalization length 14.01 days, TI 10.01 days, and BRI 2.62%. In 63 days, 85 people were hospitalized (36 M, 49 F) with a total number of 1121 days of hospitalization and an average patient age of 77.68 years (73.68 M, 80.08 F). A total of 67.05% patients had ≥3 comorbidities (24.7% had ≥5). The most common comorbidities were: neoplasia (41%), chronic renal failure (58%), D-Dimer > 500 mg/dL (50%), and COVID-19 stage (stage 3 WHO) (70%). The death rate was 27.38% (24 patients; 14 M, 10 F), patients’ average age was 86.8 years (49 M and 42 F), and the average hospitalization length was 10.39 days (min 3, max 21). Twenty-two patients that died in the MC came from RSA (91.66%) and 2 (8.34%) were unknown.

MC II Wave: average number of beds per month: 86; BOR 65.9%, average hospitalization length 15.88 days, TI 8.22 days, and BRI 3.06%. In 74 days, 264 people were hospitalized (163 M, 101 F) with a total number of 4194 days of hospitalization and an average patient age of 70.23 years (68.64 M, 72.95 F). Overall, 100% of patients had ≥3 comorbidities. The most common comorbidities were: neoplasia (35%), chronic renal failure (48%), D-Dimer > 500 mg/dL (62%), and COVID-19 stage (stage 3 WHO) (70%). The death rate was 28% (74 patients; 43 M, 31 F), patients’ average age was 82.28 (80.57 M and 84.61 F), and the average hospitalization length was 10.83 days (min 1, max 26). Thirty-four patients that died in the MC came from RSA (46%), 18 (24.32%) from home, and 22 (27%) were unknown.

Overall MC: average number of beds per month: 63.8; BOR 58.97%, average hospitalization length 15.52 days, TI 10.80 days, and BRI 5,47%. In 144 days, 350 people were hospitalized (199 M, 151 F) with a total number of 5418 days of hospitalization and an average patient age of 70.03 years (69.4 M, 75.54 F). The most common comorbidities were: neoplasia (38%), chronic renal failure (53%), D-Dimer >500 mg/dL (56%), and COVID-19 stage (stage 3 WHO) (70%). The death rate was 27.71% (98 patients; 56 M, 41 F), patients’ average age was 82.67 (81.01 M and 84.92 F), and the average hospitalization length was 10.83 days (min 1, max 26). Fifty-six patients that died in the MC came from RSA (57.14%), 18 (18.37%) from home, and 24 (24.5%) were unknown.

The results have been depicted in a Barber–Johnson nomogram (Figure 4).

Details about the ward characteristics, from a management perspective, are also summarized in Table 1.

## 4. Discussion

The group of SARS-CoV-2-related pathologies represent an unexpected and unpredictable challenge for healthcare systems worldwide. Similarly to other major public health challenges in the past, the long-term consequences will likely be the subject of meticulous future studies. This has already been the case for a wide range of public health arguments, such as asbestos toxicity [30], rural arbovirosis [31], the 1918 influenza long-term consequences [32], and the South African health system’s structural sustainability [33].

With regard to the collected data, the best strategy to decrease the COVID-19 death rate, especially in a territory characterized by the availability of several facilities dedicated to frail patient care, is to reduce admissions to hospital facilities. This can be achieved through active healthcare surveillance based on a remote health check system, such as the telemedicine services [34,35] implemented in 2018 in the MC ward and MCF ward of ASL Roma 6, and still used to monitor the clinical conditions of discharged patients.

In order to better understand the meaning of the data collected to date, it is necessary to affirm that, during the year 2020, the total number of hospitalization days in the MCF ward was 66% of the previous year’s number (6531 vs. 9168). Moreover, it is necessary to consider that the MCF ward was active for 365 days in 2019 and 250 days in 2020, compared to the MC ward, which was active for just 144 days during 2020. This is the reason that there is an evident difference among the derivative indexes (TI, BRI, and BOR).

Moreover, data regarding patients’ provenience were not always present on the ED reports, so it is possible that some admissions in either MC or MCF were not correctly registered if the patient was not conscious on hospital arrival—such cases, however, were rare.

Between the first and the second waves of the COVID-19 pandemic, a general lockdown of all non-essential health services occurred (i.e., screening plans, day hospital activities for chronic diseases, most outpatient visits, elective procedures, etc.). On the other hand, a reduction in ED total admissions was observed, likely due to the fear of infection if approaching a non-COVID-19-free structure. As a result of this, both underestimation and negligence were detected in people’s awareness of their own health condition. During the second wave, the combination of these two phenomena, still partially ongoing, coupled with a chronic lack of personnel in the Italian public and private healthcare system [36] (i.e., as long as the relaxation of the national and regional lockdown regime occurs, with the reintroduction of day-to-day activities and less social distancing) caused a decrease in the average age of the patients, in terms of both sheer hospitalization and death toll. As a matter of fact, the data analyzed by this study showed that the virus had a strong impact on frail patients or patients already affected by compromised conditions.

A further important consideration appears necessary: during the first wave, the virus mostly affected frail patients within large clusters and patients unable to independently conduct their own lives (i.e., multiple clusters occurred in RSA, retirement houses, cohousing experiences, etc.), a group obviously of a high average age. In the second wave, instead, SARS-CoV-2 spread across the local population, with a strong impact on a group of people who, in spite of their old age, were able to live their lives semi-independently.

As a matter of fact, the average age of people hospitalized during the second wave was about 10 years lower than those in the first wave. The death rate was largely the same (27.38% vs. 27.27%) in patients with similar characteristics (i.e., age, comorbidity, etc.), consistent with the findings presented in the literature [1,2,37].

The average number of hospitalization days due to COVID-19 were generally higher in the MCF (15.5 days vs. 8.5 days). The occupancy rate of beds (BOR) was almost half in the MC (90% vs. 59%) and BRI was 5.5% vs. 33%. The turnover rate was, therefore, 10.8% in the MC compared to 7.5% in the MCF.

Furthermore, during the pandemic, 228 (25%) out of 970 patients hospitalized in the MC were treated with NIV, consistent with the epidemiology of current Italian clinical procedures [38].

## 5. Conclusions

A COVID-19 patient represents a kind of patient who, independently of age and gender, needs more resources and for a longer period of time compared to the typical patient of an Internal Medicine ward. Their categorization is somewhere between Internal Medicine and TI. This is particularly evident because 25% of COVID-19 patients needed NIV (in some periods, up to 50%), intensive care, and constant monitoring. As a consequence of this, the Internal Medicine ward changed from being simply a high-complexity ward to a ward with both medium and high complexity, such as a sub-intensive therapy ward.

Unfortunately, most fatalities (57.14%) occurred in patients admitted from local facilities. For such patients, prevention is a strategy far better than treatment: the best strategies must yet rely on proven preventive measures (such as facial mask usage, fast tracing systems, the distribution of antigenic swabs for admission screening, and the use of telemedicine systems) in order to avoid hospital admission entirely, given the likelihood of death in this subgroup of patients [39].

### Implications for Professional Practice and Policy Makers

It is important to conceptually transform the Internal Medicine ward in a “projection” [40] to a sub-intensive therapy ward (according to the military NATO Joint Response Force concept). This would allow both the monitoring and treatment of pathologies linked to respiratory and cardio-circulatory systems and a reduction in bacterial super- and co-infections, particularly those occurring in hospitals and those directly related to antibiotic resistance.

The implementation of a new ward model must, necessarily, suggest the introduction of a new, functional, and architectural concept of a hospital. This new concept would provide redesigned areas, make hospitalization more “tolerable” (e.g., green areas, well-lit locations, and single rooms), and allow for a new equitable distribution of beds in the different wards. This can be made possible by decreasing the number of beds for planned hospitalizations and by increasing the number of beds for seriously ill patients.

Such an approach must, however, keep pace with recent developments in technological and IT capabilities. Since transboundary spillover of technological innovation can occur within any field and influence the energy efficiency and sectoral performance of other areas, it is desirable that tomorrow’s health systems will be designed with consideration of knowledge-sharing principles and economic and ecological sustainability [41].

## Figures and Tables

**Figure 1 ijerph-18-05999-f001:**
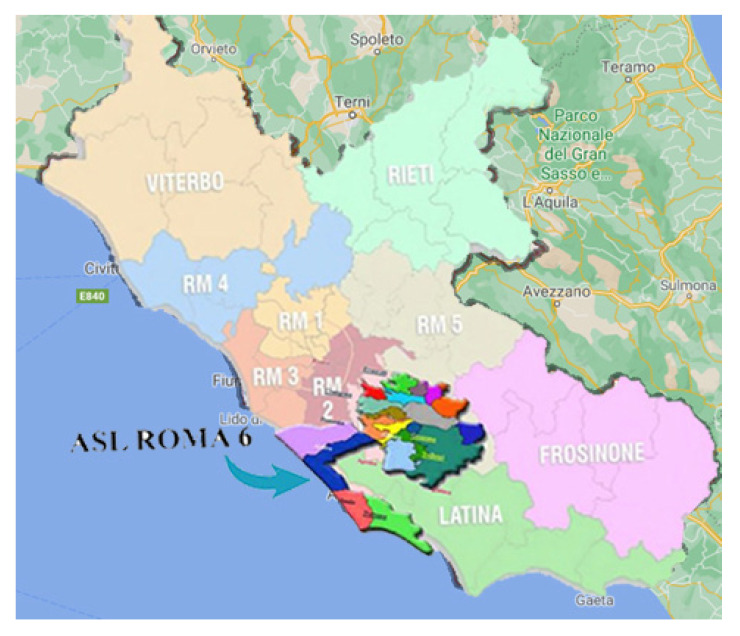
Health authority districts of Lazio region. The figure shows the geographical location of the Rome 6 ASL (in vivid colors) in the context of the Lazio Region and the relationship with the other ASLs.

**Figure 2 ijerph-18-05999-f002:**
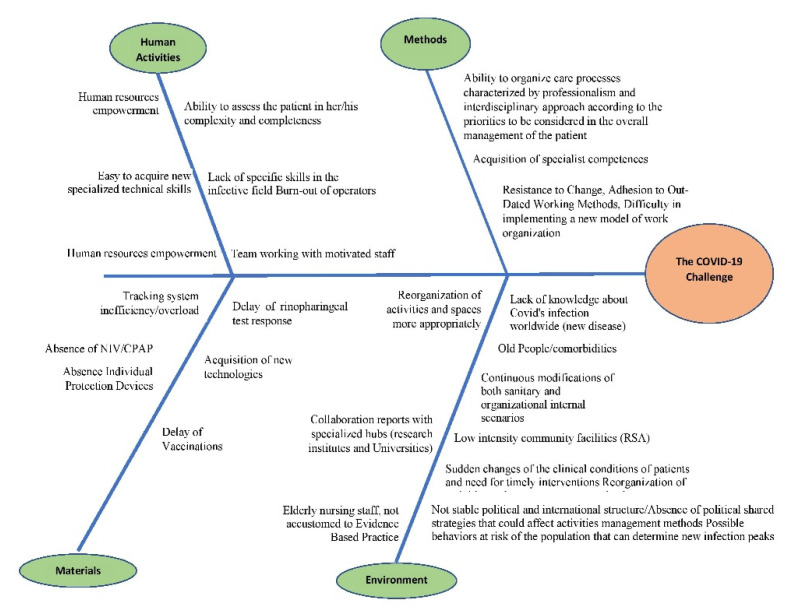
Ishikawa diagram.

**Figure 3 ijerph-18-05999-f003:**
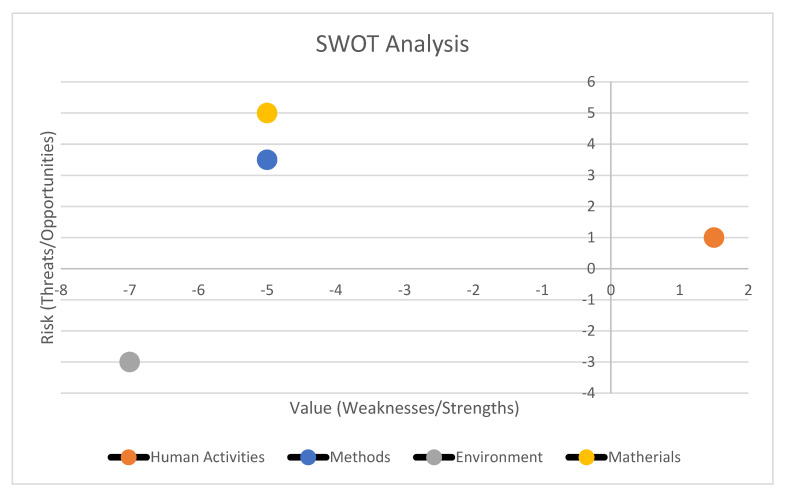
Plot graph of SWOT analysis. Comfort Zone: upper left. Danger Zone: lower right. Challenge Area: lower left. Improvement Area: upper right. The score is given on the abscissa, by the pros/cons ratio, and on the ordinate by the potential risk/perceived value.

**Figure 4 ijerph-18-05999-f004:**
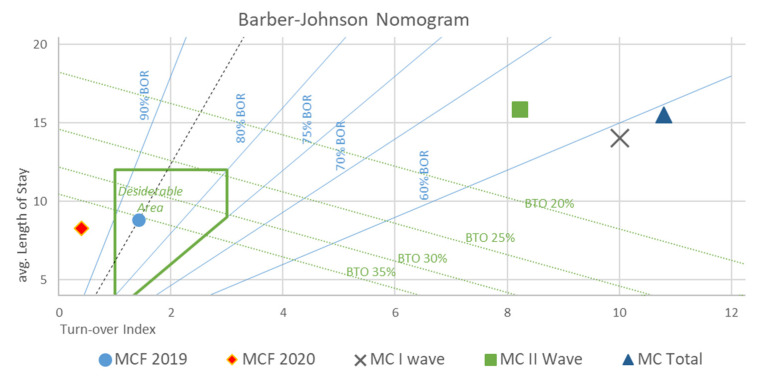
Barber–Johnson nomogram depicting ward performances during 2019–2020 period. *x*-axis: average length of stay (days); *y*-axis: turnover index (days).

**Table 1 ijerph-18-05999-t001:** Results. B/M: beds per month average value; BOR: bed occupancy rate, LOS: length of stay, TI: turnover index; BRI: bed rotation index. TOT: total number of patients; M: male, F: female, AV.AG: average patient age; AD: activity days (working days during which the ward was active); HDD: hospital delivered days (total days of assistance delivered, cumulative). MCF: Internal Medicine COVID-19free ward, MC Internal Medicine COVID-19 ward.

	B/M (BEDS)	BOR	LOS (DAYS)	TI (DAYS)	BRI	TOT (M/F) (PATIENTS)	AV.AG (M/F) (YEARS)	AD (DAYS)	HDD (DAYS)
MCF 2019	29.25	86.11%	8.8	1.42	36.5%	1038 (496/542)	79.03 (78.18/79.81)	365	9168
MCF 2020	26.5	98.98%	8.26	0.09	29.81%	791 (394/397)	77.73 (75.42/80.02)	249	6531
MC I WAVE	30.5	58.34%	14.01	10.01	2.62%	85 (36/49)	77.68 (73.68/80.8)	63	1121
MC II WAVE	86	65.90%	15.88	8.22	3.06%	264 (163/101)	70.22 (68.74/72.95)	74	4194
MC TOTAL	63.8	58.97%	15.52	10.8	5.47%	350 (199/151)	72.03 (69.39/75.54)	144	5418

## Data Availability

Raw data were generated at PO Castelli Hospital. Derived data supporting the findings of this study are available from the corresponding author F.R. on request.

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
