# Peer review of "Burden of COVID-19 on Italian Internal Medicine Wards: Delphi, SWOT, and Performance Analysis after Two Pandemic Waves in the Local Health Authority “Roma 6” Hospital Structures"

_ijerph, 2021, doi:10.3390/ijerph18115999_

Round 1
Reviewer 1 Report
Very interesting work. I propose to expand the "discussion" section and separate a sub-section with implications for professional practice. Adjust the bibliography to the editorial requirements.
Author Response
Dear Editor,
we are deeply thankful to the Reviewers for the suggestions received, and have modified the paper in order to meet the indications.
In particular, we would like to thank Rev. 2, who performed a punctual dissection of the paper and highlighted several elements that necessitated more clarity. Thanks to his suggestions, the whole paper underwent a major refactoring in its structure in order to explicitate points that would else have been overlooked.
Consequentely, several section such as Title, Abstract, Text, Bibliography and Author Contributions have undergone major rework.
Hereby follows the indication of aour interventions in response to every point made by the Reviewers.
Rev 1 - Discusison section has been expanded with more recent evidence, and an "Implications for professional practice and policy makers " sub-section has been added, according to the reveiwers suggestion.
Bibliography has been adjusted according to editorial requirements.
best wishes,
Dott. F Rosiello
Reviewer 2 Report
The manuscript aimed to assess the burden of COVID-19 on internal medicine wards considering data limited to a specific Italian local health unit – even if this last aspect is not clear in the title. There are several missing aspects and the quality of the presentation affects the soundness of the work. Some of the main sections need to be deeply revised in form and content. I indicate some of the critical aspects and suggestions for the authors.
- The Introduction does not provide sufficient background and the aim of the study should be better explained. The References are not very appropriate (please, see comment n.5 for details). Furthermore: Line 32, please indicate the meaning of the mentioned starting date, ref. WHO; Lines 34-36, the number of cases is not properly indicated (“1200.00”) and is not updated as well as the number of deaths that is sadly higher than the indicated number. More recent international and national reports should be mentioned; Lines 52-79, The paragraph related to the local health unit “Azienda ASL Roma 6” should be more connected to the introduction. All the territorial details are mostly not relevant for the study because they are not focused to detail the rationale behind the selection of that local unit as a representative case of the Italian situation; Figure 1 does not represent any additional value for the study.
- Methods: The section is not well-defined. The methodologies of the study are not clearly described and some results are reported as methods. The second stage of the study has not even clearly mentioned in the text. The third one is just mentioned briefly and unclearly presented even if the results of the entire study are focused on that part. Some other aspects: Lines 81-84: The authors probably meant to say that the aim was to analyze the general causes which may have been contributed to the patients' deaths but, at line 82, the statement should be clarified; Figures 2 and 3 report results related to the application of techniques. Thus, they should be not reported in Methods but in Results and the results should be discussed; Lines 104-105, “wards performances”: It is not really clear what identifies the performance, which parameters/indicators have been defined and compared. If the data are the ones reported at lines 114-165 and they have been elaborated from initial public data, the methodology used for the analysis, indicators/parameters, and the rationale of their selection should be described.
- Results: As previously indicated, the section does not include the results of the other stages of the study, partially indicated in methods. Furthermore: Lines 111-113, the authors stated that the results came from the analysis of SDO of patients (ASL6?, please indicate it) from a national IT system but, the original data from the database are not reported and it is not clear which are the data elaborated from the authors and the data extracted from the source and just reported in the text; Data indicated at lines 114-165 should be better reported in tables including all the parameters or revising Table1 adding all the values reported in the text (e.g., death rate, comorbidity, etc.) and the missing units; Lines 139-141 report the details of comorbidities, they should be also indicated in details for the other groups (II wave, MCF2020 ...); Lines 166-167, there is no figure of the mentioned result “Barber-Johnson’s monogram” in the text.
- Discussion: Lines 179-181: The sentence “all data evaluated…are coherent” should be clarified avoiding misleading statements. No evidence is reported supporting the statement. It is also necessary to explain and quantify how the results could be considered representative and of which context (the national – Italian or the local territory?); Lines 194-195: the difference is not quantified and it is not clear what the authors mean for “derivates indices”; Line 223: abbreviation ORB, to be checked in ref. to “BOR”-Table1.
- References: n.1 is not specific as a reference for the number of reported cases; n.4. is not appropriate as a reference to the number of deaths; n.5 and n.6 other updated reports should be considered; n.10 is not appropriate as a reference for the availability of COVID-19 vaccines in Europe, other international references should be considered; n.24, it is not a reference.
Author Response
Dear Editor,
we are deeply thankful to the Reviewers for the suggestions received, and have modified the paper in order to meet the indications.
In particular, we would like to thank Rev. 2, who performed a punctual dissection of the paper and highlighted several elements that necessitated more clarity. Thanks to his suggestions, the whole paper underwent a major refactoring in its structure in order to explicitate points that would else have been overlooked.
Consequentely, several section such as Title, Abstract, Text, Bibliography and Author Contributions have undergone major rework.
Hereby follows the indication of aour interventions in response to every point made by the Reviewers.
Rev 2 - 1) Introduction has been reworked, in the hope of giving more clarity. Numbers and references have been updated. A better description of the territory and the reasoning behind the reationale of its study have been indicated.
2) The "Materials and Methods" section has been rewritten from scratch, with detailed references. Most of the previous part has been rewritten or moved to Results section. Methodology used for analysis has been described.
3) Data sources have been explicitated. Added previously missing Figure 4.
4) Results section underwent major changes.
5) Statements have been clarified and references added. Also minor typos have been corrected.
Best wishes,
Dott. F Rosiello
Reviewer 3 Report
Now it is a hot topic for measuring strengths, weakness, opportunities and threats before and after pandemic waves for COVID-19, this paper did a good and useful work for that.
1.SWOT Analysis method is relative easy and simple method, why Delphi method and Cartesian graph are suitable for this anylysis? you should give more proof or reasons to use that.
2.The literature review part is too weak, the author has not given a relatively comprehensive review on the existing literature, and therefore it is hard to highlight the contribution of this research comparing with existing research. some fresh paper should be added:
Sun H., Edziah B K., Kporsu A.K., Sarkodie S A., Taghizadeh-Hesary F. Energy efficiency: the role of technological innovation and knowledge spillover,Technological Forecasting & Social Change,2021,120659.
3.Any data should have its source, so many data has no data source, and this is not acceptable for a Journal paper.
4.Please check the whole paper, make sure the language fulfill the requirement of the journal.
5.Conclusions, should be: Conclusions and Policy Implications, it is better to give several suggestions for policy maker, like 1, 2, or 3.
Author Response
Dear Editor,
we are deeply thankful to the Reviewers for the suggestions received, and have modified the paper in order to meet the indications.
In particular, we would like to thank Rev. 2, who performed a punctual dissection of the paper and highlighted several elements that necessitated more clarity. Thanks to his suggestions, the whole paper underwent a major refactoring in its structure in order to explicitate points that would else have been overlooked.
Consequentely, several section such as Title, Abstract, Text, Bibliography and Author Contributions have undergone major rework.
Hereby follows the indication of our interventions in response to every point made by the Reviewers.
Rev 3 - 1) The "Materials and Methods" section has been rewritten from scratch, with detailed references.
2) more recent evidence has been added, including the one suggested;
3) precise references have been added for every instance where external data was presented;
4) the paper underwent extensive editing of English language and style
5) Conclusion section has been reworked, and an "Implications for professional practice and policy makers" sub-section has been added, according to the suggestion.
Best wishes,
Dott. F. Rosiello
Round 2
Reviewer 2 Report
The manuscript has been revised according to some of the initial comments but some of them have been not considered. The quality of the presentation is improved in some parts. The content remains limited. Some revisions and improvements are still needed.
- References: The initial comment at the first stage of the revision has been probably not very clear to the authors. Self-citations included in the text with the current numeration n.10 and n.19 are not appropriate, please remove them from the manuscript. At the same time, please remove the reference n.5 (n.1 in the previous version) since is not specific as a reference for the number of reported cases; n.13 and n. 15 are not appropriate references for the sentence at lines 44-47.
- Abstract: The abstract should be revised according to the improvements performed in the text. Line 20-21: please, revise the terms “death reasons” according to the clarified form and meaning at lines 107-108; Lines 30-31: please, clarify the form of the sentence.
- Lines 126-128: the authors should specify clearly which are the data extracted from the source and just reported in the text and which are the calculated parameters/indicators and relative calculated statistics.
- Lines 205-209: as already indicated at the first stage, the details of comorbidities are reported only for “MC I Wave”. Why are they considered relevant only for that group? Otherwise, they should be also indicated in detail for the other groups.
- Table1: the units are still missing.
- Lines 246-248: as already commented at the first stage of the revision, the lines should be revised and clarified avoiding misleading statements. It is not sufficient to add just some references. Since the manuscript is supposed to be a scientific paper, a deep and pertinent discussion, supported by international publications, is needed.
- Lines 301-305: final conclusions should be limited to the results of the study.
Author Response
Dear,
1) Unfortunately, due to the major rework of the paper, some references were either misplaced in order, missing or leftovers: we are deeply sorry for this overlook on our side.
In order to solve the problem, a citation software was used (Zotero), hoping that it will not cause conflict with editorial standard recommendations.
References 10 and 19 have been removed. Reference 5 has been moved to proper place (36 in current version). References 13 and 15 have likewise been moved to proper position (10 and 31 in current version). Furhter references were also added or removed in some places. Duplicate references have been merged.
2) Abstract has been revised as suggested. Terms "death reasons" have been replaced with "bad outcome causes". Also, the sentence was phrased in a way that suggested that only one hospital structure was implicated in the study conduction, while actually there were 2 different structures with several wards in. Due to word count limitations, this is indicated in the abstract with the word "aggregate Italian COVID wards cohort" and has also been explicitated in the text in point 2.2 Population, Variables and Data Sources.
Abstract conclusions have also been rephrased.
3) Retrieved and calculated data have been explicitated and acronyms used in following instances in the text.
4) Data regarding comorbidities have been added for all COVID cohorts.
5) Table 1: Units have been added in table headers. Also, Units have been added in Fig. 4 description.
6) In order to avoid misleading, confusionary or unsupported statements in the paper, the sentence has been removed entirely.
7) Conclusions have been modified, and paragraph about prevention measures was adjusted to clarify its meaning.
best wishes,
the authors
Reviewer 3 Report
The authors have revised the paper according to the comments very well. I suggest it could be accept directly. One more point:
The topic can be shorter and more refined.
Author Response
Dear,
Thanks for the general green light. It is not clear to us, howewer, what the statement "The topic can be shorter and more refined" means. Is it referred to the title? if this is the case, please note it has been further modified: it now reads: "Burden of COVID-19 on Italian Internal Medicine ward: Delphi, SWOT and Performance analysis after two pandemic waves in the Local Health Authority “Roma 6” hospital structures".
Best wishes,
the authors